# Voltage-gated proton channels from fungi highlight role of peripheral regions in channel activation

Chang Zhao [1,2] & Francesco Tombola [1,2]✉

Here, we report the identification and characterization of the first proton channels from fungi. The fungal proteins are related to animal voltage-gated Hv channels and are conserved in both higher and lower fungi. Channels from *Basidiomycota* and *Ascomycota* appear to be evolutionally and functionally distinct. Representatives from the two phyla share several features with their animal counterparts, including structural organization and strong proton selectivity, but they differ from each other and from animal Hvs in terms of voltage range of activation, pharmacology, and pH sensitivity. The activation gate of Hv channels is believed to be contained within the transmembrane core of the protein and little is known about contributions of peripheral regions to the activation mechanism. Using a chimeragenesis approach, we find that intra- and extracellular peripheral regions are main determinants of the voltage range of activation in fungal channels, highlighting the role of these overlooked components in channel gating.

[1] Department of Physiology and Biophysics, University of California, Irvine, CA, USA. [2] Chao Family Comprehensive Cancer Center, University of California, Irvine, CA, USA. ✉email: ftombola@uci.edu

Fungi and humans have a close, yet complicated, relationship: while many gill mushrooms are great resources of proteins and some filamentous fungi are widely exploited in the food industry, some are pathogenic to plants, humans, and wildlife. Among at least 2.2 million fungal species[1], over 8000 are known to infect plants and around 300 cause human diseases[2]. Fungi are capable of adapting to fluctuating, sometimes extreme, conditions. While most thrive in mildly acidic environment (e.g., growing on decaying or fermenting substrates), pathogenic fungi can survive in host organs with a wide range of pHs (pH 2–8)[3]. Different types of fungi actively modify the surrounding proton concentration by secreting organic acids[4] or ammonium[5], promoting hyphae germination for host tissue invasion, meanwhile maintaining a relatively stable intracellular neutral pH[6,7].

All living organisms use active and passive proton transport mechanisms to control intracellular pH and proton gradients across cell membranes. Passive mechanisms, mediated by ion channels, let protons flow along their electrochemical gradient in response to specific signals. In animal cells, two major classes of proton channels have been identified: voltage-gated Hv channels[8,9] (also known as VSOPs) and otopetrins[10]. In human, the Hv1 channel contributes to pH homeostasis in various cell types and has important functions in the immune, respiratory, and reproductive systems[11], e.g., its activity is known to modulate the production of reactive oxygen species (ROS) by NADPH oxidase (NOX) enzymes[12–14]. Otopetrin 1 (Otop1) on the other hand, plays important roles in the sensory nervous system, acting as sour taste receptor[10,15], and supporting various aspects of vestibular function[16,17].

Homologs of NOX enzymes have been identified in fungi as well and are known to be critical for filament growth and for infection and penetration of the host surface[18]. Moreover, pH sensing and signaling in fungi involving the PacC/Rim pathway have been particularly associated with fungal virulence[3,19,20]. It is reasonable to assume that fungi could use passive transport mechanisms mediated by proton channels for pH regulation, in addition to the known active mechanisms mediated by the H⁺-ATPase (Pma1) on the plasma membrane[7,21,22] and the V-ATPase in intracellular vacuoles[23,24].

In this study, we report the identification of members of the Hv channel family in both higher and lower fungi and the biophysical and pharmacological characterization of two of these channels: SlHv1 from *Suillus luteus*, a representative of the phylum *Basidiomycota*, and AoHv1 from *Aspergillus oryzae*, a representative of the phylum *Ascomycota*. We find that fungal Hvs share several features with their animal counterparts, including strong proton selectivity and gating modulation by transmembrane pH gradient (ΔpH), but the channels differ from each other and from animal Hvs in terms of voltage range of activation, pharmacology, and pH sensitivity in the absence of transmembrane ΔpH, which suggest functional adaptation to different environments.

Animal Hv proteins consist of an amphipathic helix S0 and four transmembrane helices S1 through S4 which form a voltage-sensing domain (VSD) structurally equivalent to the VSDs of voltage-gated Na⁺, K⁺, and Ca²⁺ channels[8,25,26]. The VSD of Hv channels contains the H⁺ conduction pathway, whereas a distinct pore domain, linked to the VSD, contains the ion conduction pathway in other channels[27–29]. Another structural feature typical of Hv proteins is a cytoplasmic coiled-coil domain (CCD) that mediates dimerization and is connected to the S4 helix of the VSD[30–32].

Our current understanding of the mechanism of activation of Hv channels is based on studies focused on the S1–S4 transmembrane region, as the activation gate is thought to be located in this part of the protein[33–37]. Here, we find that SlHv1 and AoHv1 share the same structural elements found in animal Hvs.

We then use a chimeragenesis approach to identify protein regions responsible for the strong difference in voltage-dependent activation between the two fungal channels. Our result point to previously unrecognized roles of peripheral regions—defined as portions of the protein interacting with the membrane surface, including loops connecting the transmembrane helices—in the activation process.

## Results

**Identification of putative proton channels in Fungi**. The importance of pH regulation in fungi raised the question of whether these organisms possess proton channels similar to those found in the animal kingdom. Through BLAST search, we identified a group of putative Hv channels from the following organisms: *Hypsizygus marmoreus*, *Amanita muscaria*, *Psilocybe cyanescens*, *Suillus luteus*, *Scleroderma citrinum*, *Galerina marginata*, *Mycena chlorophos*, *Agaricus bisporus*, *Piriformospora indica*, *Fusarium oxysporum*, *Sclerotinia sclerotiorum*, *Cladophialophora immunda*, *Talaromyces marneffei*, *Penicillium brasilianum*, *Aspergillus oryzae*, and *Aspergillus flavus* (see "Methods" section for NCBI sequence IDs). We also searched for Otop orthologues in fungi but were unable to find any fungal protein related to this other type of proton channels (see "Methods" for details).

Overall, the putative fungal Hvs share 20–29% sequence identity with the human voltage-gated proton channel hHv1. Cladogram of proton channels from these species reveals that they are only distantly related to known Hvs, and there is a clear separation between *Fungi* and *Animalia* (Fig. 1a). Representatives from mammals, reptiles, amphibians, birds, fish, and ascidians were included in the cladogram, together with representatives from arthropods and molluscs (for a detailed comparison of animal Hvs, see ref. [38]). Phylogenetic analysis (Supplementary Fig. 1) indicates the existence of Hv channels in all five major divisions of the *Fungi* kingdom and that Hvs from slime molds (protists) are more closely related to animal Hvs than their fungal counterparts. Hv representatives from gill mushrooms and molds seem to form two distinct groups. Therefore, we selected one candidate from each group for further investigation: Hv1 from *Suillus luteus* (SlHv1) and Hv1 from *Aspergillus oryzae* (AoHv1), which share 25.7% sequence identity. Protein sequence analysis (see "Methods") indicates a similar membrane topology and structural organization for SlHv1 and AoHv1 compared to animal Hvs (Fig. 1b). The S4 helix of the fungal channels carries positively charged residues typical of other voltage sensors. However, the S4 signature motif for mammalian Hvs is R·WR··R··N (where · is usually a hydrophobic residue), while the motifs for SlHv1 and AoHv1 are R·WR··K··G and R·WR··K··E, respectively. In addition, the predicted CCD of AoHv1 is significantly shorter than the corresponding domain of SlHv1 and it is coupled directly to S4 without the linker region normally found in other Hvs (Fig. 1b, and Supplementary Fig. 2).

**Channel expression and voltage dependence of activation**. Having a structural organization similar to animal Hvs is not sufficient to predict proton channel activity, as previously shown with HVRP1/TMEM266, a membrane protein closely related to human Hv1 that does not function as a channel[39–41]. So, we expressed the fungal proteins in *Xenopus* oocyte and performed electrophysiological measurements in excised membrane patches (Fig. 1c–f). We were able to record robust voltage-dependent currents from both SlHv1 and AoHv1 using ionic conditions established for animal Hvs[28] with both intra- and extracellular media buffered at pH 6.0. Interestingly, the two proteins showed very different voltage ranges of activation and kinetic properties,

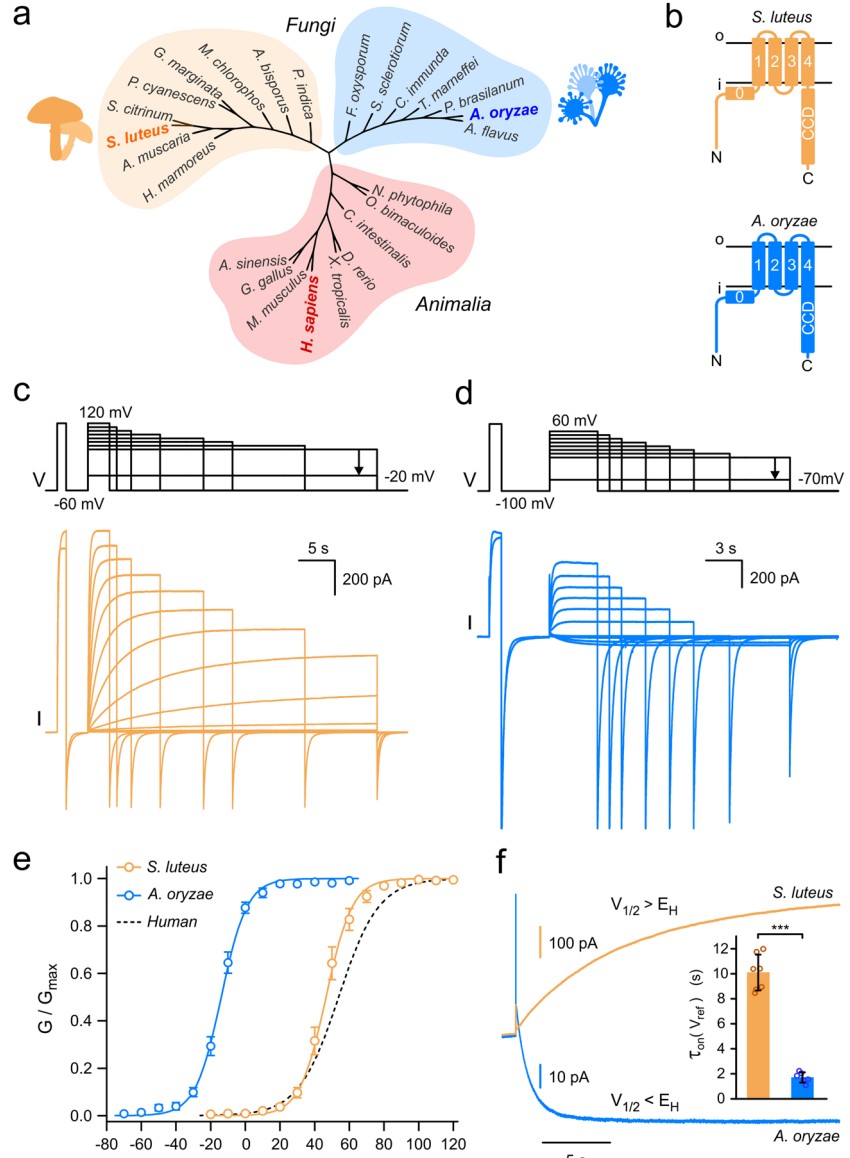

**Fig. 1 Fungal Hvs from species *S. luteus* and *A. oryzae* show distinct biophysical properties. a** Unrooted cladogram places Hv channels of fungi into two large groups distinct from animal orthologs. **b** Predicted topology of Hvs from *S. luteus* and *A. oryzae* showing connections between C-terminal coiled-coil domains (CCD) and S4 transmembrane segment of different lengths. **c, d** Representative proton currents from fungal Hvs shown in (**b**) expressed in *Xenopus* oocytes and measured from inside-out patches (pH$_i$ = pH$_o$ = 6.0). For clarity, only the first and last traces elicited by the depolarization pre-step are shown. **e** Conductance versus voltage relationships ($G$–$V$s) of SlHv1 and AoHv1 calculated from current traces like those shown in (**c, d**). Curves are Boltzmann fits. For SlHv1, $V_{1/2}$ = 46.5 ± 2.3 mV, slope = 7.6 ± 0.5 mV; for AoHv1, $V_{1/2}$ = −13.9 ± 1.1 mV, slope = 7.4 ± 0.9 mV ($n$ = 5). Error bars are SEM. $G$–$V$ for hHv1 is shown as dashed line for reference ($V_{1/2}$ = 53 ± 3 mV, slope = 11.6 ± 0.6 mV, from ref. [28]). **f** Representative activation currents of SlHv1 and AoHv1, each measured at a reference voltage ($V_{ref}$) closest to their individual $V_{1/2}$ ($V_{ref}$ = 50 mV for SlHv1 and −10 mV for AoHv1). At $V_{ref}$, $G/G_{max}$ is 0.61 and 0.63 for SlHv1 and AoHv1, respectively. $E_H$ indicates Nernst potential for protons, which in this case is 0 mV. Time constants of activation currents ($\tau_{on}(V_{ref})$) are shown in inset as mean values ± SEM ($n$ = 7 for SlHv1; $n$ = 6 for AoHv1). Welch's $t$-test was used for statistical analysis, ***$p$ < 0.001.

with SlHv1 activating slowly, and at positive membrane potentials, and AoHv1 activating rapidly, and at negative membrane potentials (Fig. 1c–e). The conductance vs. voltage relationship ($G$–$V$) of SlHv1 was slightly left-shifted compared to the $G$–$V$ of hHv1 (Fig. 1e, $V_{1/2}$ = 46.5 ± 2.3 mV for SlHv1, $V_{1/2}$ = 53 ± 3 mV for hHv1[28]), whereas the $G$–$V$ of AoHv1 was strongly left-shifted (Fig. 1e, $V_{1/2}$ = −13.9 ± 1.1 mV).

The two fungal channels also differed in their ability to conduct inward current, which depends on the relationship between the voltage range of activation and the Nernst potential for protons ($E_H$). SlHv1 opens under electrochemical gradients that favor proton efflux, with $V_{1/2}$ > $E_H$ (Fig. 1f, $I(V_{1/2})$ > 0), similar to what

is observed with the large majority of Hvs from the animal kingdom[11]. AoHv1, on the other hand, opens when the electrochemical gradient favor proton influx, with $V_{1/2}$ < $E_H$ (Fig. 1f, $I(V_{1/2})$ < 0). Besides being opposite in sign, the currents from SlHv1 and AoHv1 measured at a reference voltage $V_{ref} \approx V_{1/2}$ reached steady-state level at different rates. To quantify the difference, we fitted the currents with a single-exponential function. From the comparison of the relative time constants $\tau_{on}(V_{ref})$, we concluded that AoHv1 is approximately sixfold faster than SlHv1 (Fig. 1f, bar graph). As previously observed with animal Hvs[11], the time course of activation of SlHv1 and AoHv1 showed a time lag between the beginning of the

depolarization and the rising phase of the current (Supplementary Fig. 3a-b), suggesting that transitions between multiple closed states take place before the opening transition. The initial lag phase was more than one order of magnitude shorter than $\tau_{on}$ and was excluded from our exponential fits of the currents (Supplementary Fig. 3a–b).

Animal Hvs are homodimers[27,28,30] in which the two subunits gate cooperatively[32,42,43]. Earlier studies found that the CCD plays a critical role in Hv dimerization[27,28] and that the lag phase in the time course of activation is a characteristic of the dimeric state as it disappears in monomerized channels[44]. The presence of the CCD in fungal Hvs and the lag phase in their currents suggest that these channels could be also made of multiple subunits. To further investigate this possibility, we estimated the gating charge associated with SlHv1 and AoHv1 activation ($z_g$) using the limiting slope method[45,46]. We found that $z_g$ was ~5 for both channels (Supplementary Fig. 3c-e). Similar values were previously obtained for animal Hvs[32,43] Based on the number of positively charged residues located in the S4 helix of both animal and fungal Hvs, each subunit is expected to contribute up to 3 gating charges to the activation process. Hence, a $z_g > 3$ indicates that more than one subunit is involved in cooperative activation.

**Proton selectivity of fungal Hvs**. It is well established that Hv channels from the animal kingdom have almost perfect proton selectivity[8,9,11]. So, we set out to determine whether SlHv1 and AoHv1 share the same characteristic. We measured the current reversal potential ($V_{rev}$) under different transmembrane pH gradients ($\Delta pH = pH_o - pH_i$), and compared it to the corresponding $E_H$ (Fig. 2). Currents were recorded from inside-out patches at different membrane potentials after a pre-depolarization step to 100 mV to open the channels (Fig. 2a, c). The intercept on the $V$ axis was then plotted as a function of $\Delta pH$ (Fig. 2b, d) and compared to the relationship $E_H = -58.9\Delta pH$, describing perfect proton selectivity (see "Methods"). Under the tested conditions, both SlHv1 and AoHv1 behaved similarly to animal Hvs suggesting that the mechanism of proton selection is conserved between animal and fungal Hvs. This finding is in agreement with the presence of a highly conserved aspartate at the center of the S1 helix in all fungal Hvs. That residue corresponds to D112 in hHv1 (Supplementary Fig. 2a), which is known to be part of the proton selectivity filter[40,47–49].

**pH dependence of channel gating**. A ubiquitous feature of animal Hv channels is that their voltage dependence of activation shifts around 40 mV per unit of $\Delta pH$[11]. This property is often linked to the inability of most Hv channels to allow proton influx upon activation, because changing $\Delta pH$ to values that would favor proton influx also shifts the voltage dependence of activation to more depolarized potentials, making the channel more difficult to open. Because AoHv1 was found to allow robust proton influx (Fig. 1d, f), we asked whether the voltage range of activation of fungal channels shifts of 40 mV/$\Delta pH$ unit like in other known Hvs.

As commonly done with animal Hvs, we measured current vs. voltage (I–V) relationships under different $\Delta pH$ conditions for SlHv1 (Fig. 3a, b) and AoHv1 (Fig. 3e, f). We changed $\Delta pH$ by one positive or negative unit ($\Delta\Delta pH = 1$ or $-1$, respectively), by perfusing intracellular solutions at different pH. We observed shifts in the threshold of activation that far exceeded the expected 40 mV per $\Delta pH$ unit. The downward vertical arrows in Fig. 3a, b and Fig. 3e, f indicate the $V_{threshold}$ (defined as the voltage at which proton current is first observed) that would be expected if the fungal channels followed the general relationship between $V_{threshold}$ and $\Delta pH$ previously derived from 15 different cell types

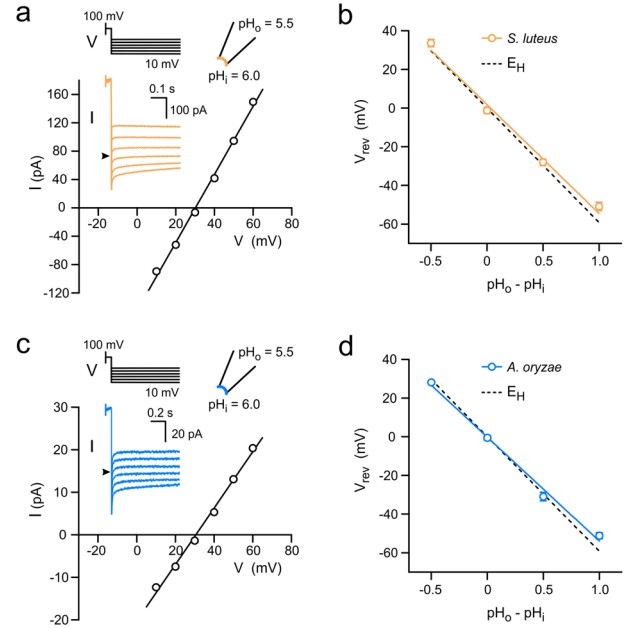

**Fig. 2 Hv1s from *S. luteus* and *A. oryzae* are proton selective. a** Example of measurement of reversal potentials ($V_{rev}$) for SlHv1-mediated currents in the presence of a transmembrane pH gradient ($\Delta pH = pH_o - pH_i$) of 0.5. Currents were measured at the indicated voltages after a depolarization step to 100 mV. Black arrowhead in inset indicates 0 pA. **b** $V_{rev}$ as a function of $\Delta pH$; slope of linear fit: $-56 \pm 4$ mV/pH unit. $E_H$, displayed in dashed line, is the Nernst potential for protons and indicates perfect proton selectivity (slope: $-58.9$ mV/pH unit). **c** Same as (**a**) but for AoHv1. **d** Same as (**b**) but for AoHv1; slope of linear fit: $-54 \pm 3$ mV/pH unit. $\Delta pH$ of $-0.5$, 0, 0.5, and 1, in (**b**) and (**d**) correspond to the following ($pH_i$, $pH_o$) pairs: (6.5, 6.0), (6.0, 6.0), (6.0, 6.5), and (5.5, 6.5), respectively. Each point in (**b**) and (**d**) represents the average of 3–4 independent measurements ± SEM. Error bars are not shown where smaller than symbols.

expressing animal Hvs[50]. Both SlHv1 and AoHv1 deviate significantly from the expected 40 mV shift per $\Delta pH$ unit. But, SlHv1 carried outward currents under all tested conditions, whereas AoHv1 carried massive inward currents when $pH_i$ was lower than $pH_o$. Its voltage dependence of activation was shifted to such hyperpolarized potentials that closing the channel completely became a challenge (Fig. 3e, gray trace).

To better quantify the shifts in voltage dependence of activation caused by changes in pH gradient, normalized $G–V$ curves were derived from I–V curves (see "Methods") (Fig. 3c, d and g, h) and the relative shifts in $V_{1/2}$ were plotted as a function of $\Delta\Delta pH$ (Fig. 3i). The resulting $\Delta V_{1/2}$ were in the order of 80–90 mV per $\Delta pH$ unit. We also measured $V_{1/2}$ as a function of pH under conditions in which $pH_i = pH_o$ (Fig. 3j, k), expecting no change as long as $\Delta pH$ remained constant, as previously observed with most animal Hvs[11]. AoHv1 did meet this expectation, with little change in $V_{1/2}$ within the 5.5–6.5 pH range (Fig. 3k). However, SlHv1 showed a clear pH dependence within the same pH range (Fig. 3j), with a $\Delta V_{1/2}$ of ~18 mV per pH unit.

**Mechanosensitivity**. Mechanical stimulation has been shown to facilitate activation of human Hv1[51]. A rise in membrane tension increases both the amount of steady-state current generated by membrane depolarization (potentiation) and the rate of activation (acceleration). Once the channel has been mechanically stimulated it remains in a facilitated state for several minutes[51]. A simple two-pulse protocol can be used to assess both potentiation

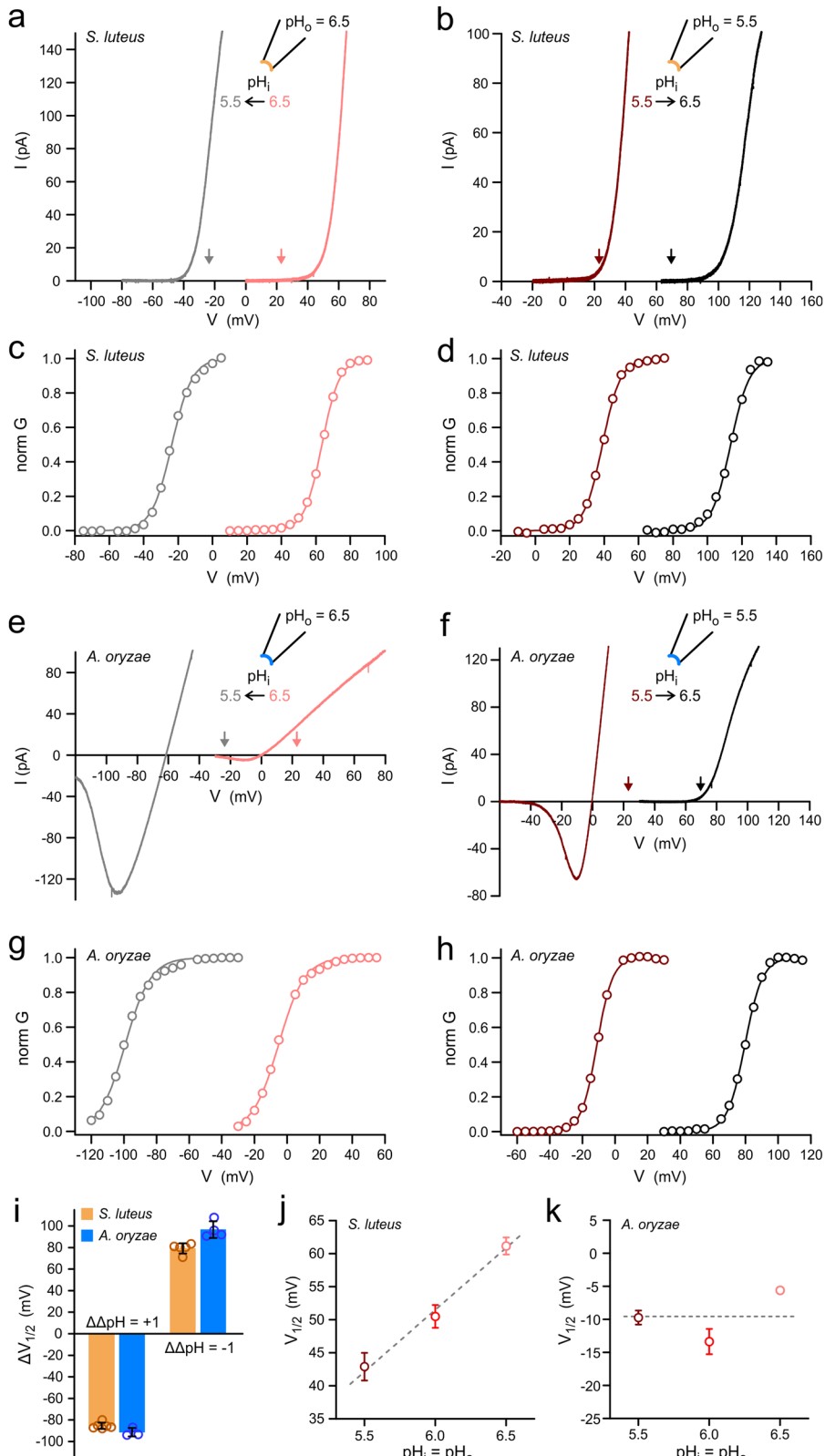

and acceleration. In this protocol, membrane tension is transiently increased between the two depolarization pulses by transient application of negative pressure to the patch pipette using a high-speed pressure clamp (HSPC). The current elicited by the depolarizing pulse that follows the mechanical stimulus can then be compared with the current elicited by the control pulse that precedes the increase in membrane tension (Supplementary

Fig. 4). We applied this protocol to both SlHv1 (Supplementary Fig. 4a) and AoHv1 (Supplementary Fig. 4b) and compared their behavior to hHv1. We found that potentiation was larger in SlHv1 compared to both AoHv1 and hHv1 (Supplementary Fig. 4c), while acceleration was similar in the two fungal channels and smaller than the acceleration in hHv1 (Supplementary Fig. 4c). Of the compared channels, AoHv1 was the least affected

**Fig. 3 Gating of fungal Hv1s is strongly ΔpH dependent. a** Change in $I$–$V$ relationship for SlHv1 in response to change in ΔpH from 0 to 1. **b** Change in $I$–$V$ relationship for SlHv1 in response to change in ΔpH from 0 to −1. Currents in **a** and **b** were measured at the indicated ΔpH in inside-out patches. Voltage was changed using ramp protocols described in "Methods". Arrows represent the voltages at which the SlHv1 should start conducting measurable current ($V_{threshold}$) if the channel followed the general behavior of animal Hv1 channels (see main text), with shifts around 40 mV per pH unit. The colors of the arrows reflect the corresponding ΔpH conditions. **c, d** $G$–$V$ relationships derived from $I$–$V$ curves in (**a**) and (**b**), respectively (see "Methods"). **e, f** same as (**a**) and (**b**) but for AoHv1. **g, h** $G$–$V$ relationships derived from $I$–$V$ curves in (**e**) and (**f**), respectively. **i** Average shifts in $V_{1/2}$ as a function of change in ΔpH (ΔΔpH) measured from $G$–$V$s, like the ones shown in (**c, d**) and (**g, h**). Each bar represents the mean of 3–7 independent measurements ± SEM. **j** $V_{1/2}$ of $G$–$V$s from SlHv1 as a function of pH under symmetrical conditions (ΔpH = 0). Each point represents the mean of 5–6 independent measurements. Error bars are SEM. Dashed line is the linear fit of the data with slope = 19 ± 2 mV/pH unit. **k** Same as (**j**) but for AoHv1. Each point represents the mean of 3–4 independent measurements. Error bars (SEM) are not shown where smaller than symbols. Data are consistent with insensitivity to pH when ΔpH = 0, shown as dashed line.

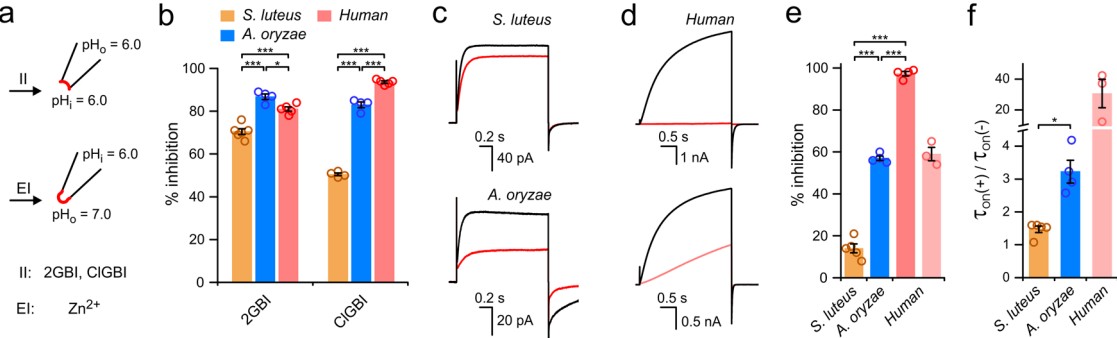

**Fig. 4 Pharmacological characteristics of Hv1s from *S. luteus* and *A. oryzae*. a** Schematics of conditions used to test intracellular and extracellular Hv1 inhibitors (II and EI, respectively) 2GBI and ClGBI were perfused on inside-out patches. $Zn^{2+}$ was perfused on outside-out patches. **b** Inhibition of AoHv1 and SlHv1 by 200 μM 2GBI and 20 μM ClGBI. Currents were measured in response to depolarization steps to 80 mV, $pH_i = pH_o = 6.0$ (see "Methods"). Each bar represents the average inhibition from at least 4 independent measurements ± SEM. **c** Representative current traces for SlHv1 (top) and AoHv1 (bottom) in response to voltage steps to 60 mV in the absence (black) or presence (red) of 100 μM $ZnCl_2$ in the bath solution. $pH_i = 6.0$, $pH_o = 7.0$. **d** Representative current traces of hHv1 measured in response to voltage steps to 60 mV in the absence (black) or presence of either 100 μM $ZnCl_2$ (red trace, top panel) or 0.5 μM $ZnCl_2$ (pale-red trace, bottom panel) in the bath solution. $pH_i = 6.0$, $pH_o = 7.0$. **e** Quantification of the inhibition of the indicated channels by 100 μM $Zn^{2+}$ (conditions as in **c** and **d**). Paler-red bar refers to the effect of 0.5 μM $Zn^{2+}$ on hHv1. Each bar represents the average inhibition from 3 to 5 independent measurements ± SEM. **f** Changes in activation kinetics induced by $Zn^{2+}$ for the indicated channels. $\tau_{on}$ values were derived from single-exponential fits of current traces in the absence (−) and presence (+) of the inhibitor (conditions as in **c** and **d**). Each bar represents the average $\tau_{on}(+)/\tau_{on}(-)$ ratio from at least 3 independent measurements ± SEM. The data for SlHv1 and AoHv1 refer to 100 μM $Zn^{2+}$. The data for hHv1 refer to 0.5 μM $Zn^{2+}$. A one-way ANOVA with Tukey's post hoc test was used for statistical analysis: *$p < 0.05$, **$p < 0.01$, ***$p < 0.001$.

by the increase in membrane tension, which is in agreement with its voltage dependence and kinetics of activation. The channel opens readily even in the absence of the mechanical stimulus, making its activation more difficult to further facilitate.

**Pharmacology.** Human Hv1 is a potential pharmacological target for the treatment of a variety of diseases[33,52–55]. Several compounds that can inhibit the channel have been identified. These include guanidine derivatives 2GBI and ClGBI, which have an intracellular binding site[56,57], and zinc ions, which bind the channel from the extracellular side[8,9,58–60]. We tested these inhibitors on fungal Hvs because they have a broad spectrum, i.e., they work on human Hv1 as well as on Hvs from other animal species. All compounds were delivered by perfusion of the bath solution during patch-clamp recordings (Fig. 4a). 2GBI and ClGBI were tested at concentrations of 200 and 20 μM, respectively, on inside-out patches with both intra- and extracellular pH at 6.0. $Zn^{2+}$ was tested at a concentration of 100 μM on outside-out patches. In this case, the extracellular pH was raised to 7.0 because less acidic conditions were previously shown to strengthen $Zn^{2+}$ binding to animal Hvs[58]. At the concentrations tested, all inhibitors reduced the proton current from hHv1 of at least 80%.

Both fungal Hvs were substantially inhibited by 2GBI and ClGBI, but AoHv1 was inhibited more effectively by the two compounds (Fig. 4b). SlHv1 was inhibited less than hHv1,

particularly by ClGBI (~50% vs. >90% inhibition, respectively). The effects of the inhibitors on AoHv1, on the other hand, were similar to those observed with hHv1 (Fig. 4b). $Zn^{2+}$ inhibited both fungal channels very poorly (Fig. 4c, d). While the human channel was inhibited almost 100%, AoHv1 and SlHv1 were inhibited <60% and 20%, respectively (Fig. 4e). The $Zn^{2+}$ concentration needed to be lowered to 0.5 μM in order for the extent of inhibition of hHv1 to be comparable with the extent of inhibition of the fungal channels (Fig. 4d, lower traces, and Fig. 4e).

2GBI and ClGBI are known to inhibit hHv1 by binding the open channel and preventing proton flow[33,56]. Conversely, $Zn^{2+}$ inhibits animal Hvs by binding preferentially the closed channel and making it more difficult to open[58]. As a result, the rate of channel opening is unaffected by 2GBI and ClGBI, but it is strongly reduced by $Zn^{2+}$. The slower opening of hHv1 in the presence of $Zn^{2+}$ can be observed in the lower panel of Fig. 4d. To assess the effect of $Zn^{2+}$ on the kinetics of channel opening in the fungal and human Hvs, the proton currents measured before (−) and after addition of the inhibitor (+) were fitted by single-exponential functions. The time constants from the fits were then used to determine the ratios $\tau_{on}(+)/\tau_{on}(-)$ (Fig. 4f). Ratios higher than 1 indicate $Zn^{2+}$-induced deceleration of channel opening.

The deceleration was somewhat more pronounced in AoHv1 compared to SlHv1 (Fig. 4f). Fungal and human Hvs could not be

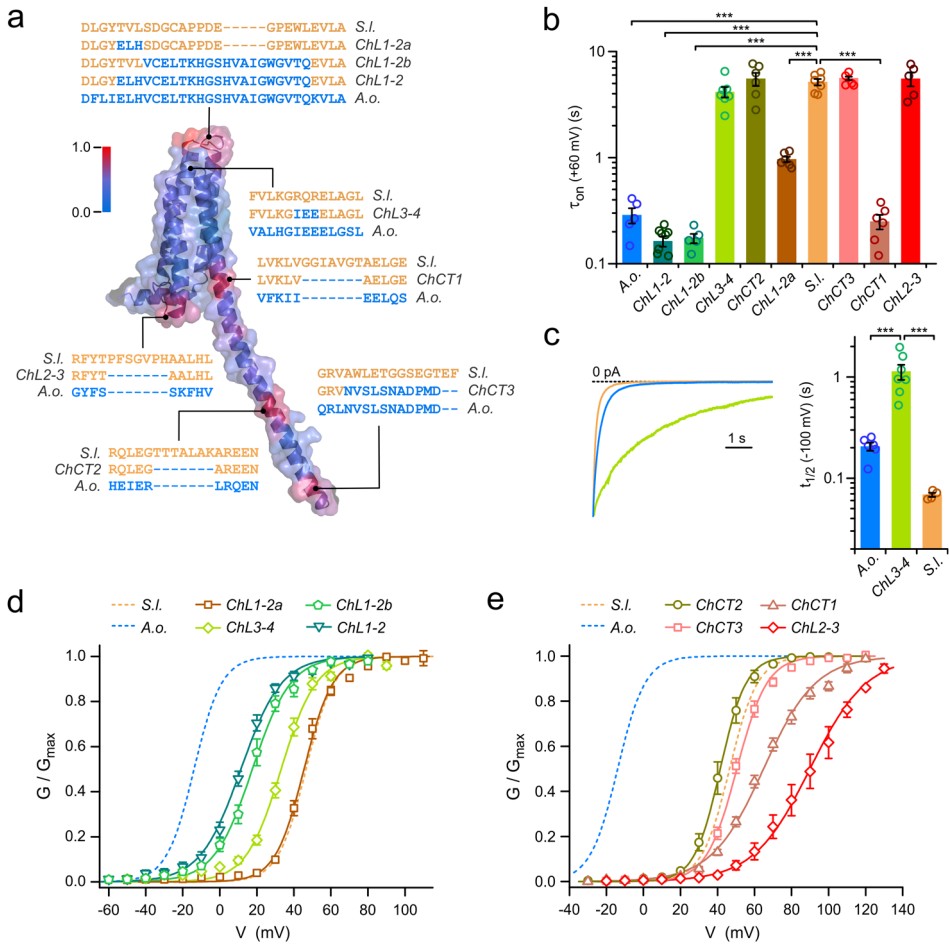

**Fig. 5 Swapping parts of SlHv1 with AoHv1 reveals regions with an important role in channel activation. a** Structural model of SlHv1 (a.a. 25–221) based on crystal structures 3WKV and 3VMX. Color of surface and cartoon representations indicates divergence in sequence homology between SlHv1 and AoHv1. Gradient varies from minimal divergence (blue) to maximal divergence (red) (see "Methods" for details). Swapped sequences in the eight SlHv1-AoHv1 chimeras: ChL1-2, ChL1-2a, ChL1-2b, ChL2-3, ChL3-4, ChCT1, ChCT2, and ChCT3 are shown with their positions within the channel structure. **b** Bar graph compares kinetics of activation of chimeric and wild type proteins. Proton currents from the indicated channels were measured in response to membrane depolarization to 60 mV and fitted with a single-exponential function with time constant $\tau_{on}$. Bars are means ± SEM ($n = 5$–9). A one-way ANOVA with Tukey's post hoc test was used for statistical analysis: ***$p < 0.001$. Only comparison with SlHv1 are shown. **c** Chimera ChL3-4 shows significant slowdown of deactivation compared to SlHv1 and AoHv1. Representative tail currents for ChL3-4, SlHv1, and AoHv1 measured at −100 mV after channel activation. Bar graph quantifies half deactivation times ($t_{1/2}$) for the three channels. A one-way ANOVA with Tukey's post hoc test was used for statistical analysis: ***$p < 0.001$. **d** Voltage dependences of chimeras with swapped extracellular regions compared to references SlHv1 and AoHv1. Each G–V relationship represents the mean of 5 to 9 independent measurements. Error bars are SEM. The following G–V parameters were derived from Boltzmann fits of the data: $V_{1/2} = 45.5 \pm 1.3$ mV, slope $= 7.8 \pm 0.3$ mV for Ch1-2a ($n = 6$), $V_{1/2} = 17.6 \pm 2.0$ mV, slope $= 10.3 \pm 0.7$ mV for ChL1-2b ($n = 5$), $V_{1/2} = 11.4 \pm 1.4$ mV, slope $= 11.0 \pm 1.1$ mV for ChL1-2 ($n = 9$), $V_{1/2} = 33.1 \pm 1.4$ mV, slope $= 9.4 \pm 0.6$ mV for ChL3-4 ($n = 7$). **e** Voltage dependences of chimeras with swapped intracellular regions compared to reference SlHv1 and AoHv1. Each G–V relationship represents the mean of 5–6 independent measurements. Error bars are SEM. The following G–V parameters were derived from Boltzmann fits of the data: $V_{1/2} = 64.9 \pm 1.7$ mV, slope $= 17.1 \pm 0.8$ mV for ChCT1 ($n = 6$), $V_{1/2} = 41.5 \pm 2.1$ mV, slope $= 7.0 \pm 0.4$ mV for ChCT2 ($n = 6$), $V_{1/2} = 50.5 \pm 1.2$ mV, slope $= 8.2 \pm 0.4$ mV for ChCT3 ($n = 6$), $V_{1/2} = 90.0 \pm 3.9$ mV, slope $= 15.3 \pm 1.7$ mV for ChL2-3 ($n = 5$). All measurements were performed at $pH_i = pH_o = 6.0$.

compared at 100 μM $Zn^{2+}$ because the remaining currents for hHv1 at this concentration were too small to fit. Nonetheless, the deceleration observed with hHv1 at 0.5 μM $Zn^{2+}$ was an order of magnitude higher than the deceleration observed with AoHv1 at 100 μM $Zn^{2+}$, (Fig. 4f). Considering that 0.5 μM and 100 μM $Zn^{2+}$ produce similar levels of inhibition in hHv1 and AoHv1, respectively (Fig. 4e), the large difference in deceleration of the opening process suggests that $Zn^{2+}$ interacts with fungal Hvs in a way that is distinct from the interaction with hHv1.

**Role of intra- and extracellular regions in gating modulation.** Surprised by the large difference in kinetics and voltage

dependences of activation between the two fungal Hvs, we wondered whether amino acid sequence comparison, guided by structural information from homology modeling, could point to divergent regions in the two proteins responsible for the different functional properties. We constructed a homology model of SlHv1, based on the available crystal structures of mHv1cc[25], a chimera between mouse Hv1 and the voltage-sensitive phosphatase CiVSP, and the isolated CCD from mouse Hv1[32]. Figure 5a shows the model representing one subunit of the homodimeric channel. From a sequence alignment of SlHv1 and AoHv1, we derived a scale of sequence divergence, defined as deviation from average similarity (see "Methods"), and converted

it into a color gradient scale (blue to red), which we then mapped on the SlHv1 homology model. To assess the extent to which spatial patterns of high divergence on the map depends on the modeling template, we mapped the scale on an additional homology model based on the VSD structure of CiVSP[61], Supplementary Fig. 5.

The core of the channel, formed by transmembrane helices S1 through S4, was the most conserved part of the two proteins in both models. In contrast, multiple intracellular and extracellular peripheral regions showed hotspots of sequence divergence (Fig. 5a and Supplementary Fig. 5), including the S1–S2, and S3–S4 extracellular loops, the S2–S3 intracellular loop, the terminal region of S4 connecting the VSD to the CCD, and the central and C-terminal parts of the CCD (the model based on CiVSP was limited to the VSD as the original protein does not contain a CCD). A comparison between the two homology models showed local differences in all the divergent regions and, in particular, in the region containing the S2–S3 loop (Supplementary Fig. 5). However, the overall pattern of divergent regions was the same in the two models.

Since AoHv1 has functional properties that set it apart from other known Hvs, we tested whether replacing any individual divergent region of SlHv1 with the corresponding region of AoHv1 could transfer some of these unique properties to the resulting chimeric channel. We generated eight such chimeras, ChL1-2, ChL1-2a, ChL1-2b, ChL2-3, ChL3-4, ChCT1, ChCT2, and ChCT3 (Fig. 5a) and compared the kinetic properties of their currents and voltage dependence of activation ($G$–$V$s) to those of the parent proteins AoHv1 and SlHv1.

We found that multiple chimeras had accelerated activation kinetics with $\tau_{on}$ values similar to AoHv1 (Fig. 5b), these included the channels in which the swapped regions were in the S1–S2 loop (ChL1-2, ChL1-2a, ChL1-2b) or in the S4-CCD linker (ChCT1). On the other hand, none of the chimeras displayed slower activation compared to SlHv1 (Fig. 5b). Because the deactivation kinetics of Hv channels have more than one exponential component[62], we measured the half deactivation time ($t_{1/2}$, see "Methods") to simplify the comparison between chimeras and parent proteins. For most chimeras, $t_{1/2}$ values were either similar to SlHv1 or smaller (faster deactivation). A notable exception was ChL3-4, in which a swap within the S3–S4 loop produced a dramatic increase in $t_{1/2}$, indicating a deactivation much slower than the deactivation of both SlHv1 and AoHv1 (Fig. 5c).

Most of the channels with chimeric extracellular regions exhibited $G$–$V$ curves shifted to more negative potentials compared to SlHv1 (Fig. 5d), whereas most channels with chimeric intracellular regions had $G$–$V$ shifted to more positive potentials (Fig. 5e). The extracellular region between S1 and S2 was the most effective at transferring AoHv1 properties to SlHv1. The $G$–$V$ curve of ChL1-2 was shifted $-35$ mV compared to SlHv1, bringing the voltage dependence of activation closer to AoHv1 than SlHv1 (Fig. 5d) mostly via a strong acceleration of channel opening (Fig. 5b). In ChL1-2, both the outermost part of the S1 helix and the S1–S2 loop are AoHv1. To determine which of these two structural components was responsible for the AoHv1-like properties, we swapped them individually in chimeras ChL1-2a and ChL1-2b. ChL1-2a was accelerated compared to SlHv1 (Fig. 5b), but its voltage dependence of activation was the same as SlHv1 (Fig. 5d). ChL1-2b, on the other hand, was more accelerated (Fig. 5b) and its $G$–$V$ curve was shifted $-29$ mV compared to SlHv1, pointing to the region containing the S1–S2 loop as the component with the largest contribution to the activation properties of the chimera. The $G$–$V$ curve of the ChL3-4 chimera was also shifted to more negative potentials compared to SlHv1 (Fig. 5d, $\Delta V_{1/2} = -13$ mV). But the shift was caused by a deceleration of channel closing rather than an acceleration of channel opening (Fig. 5b, c).

Because the opening and closing processes were differentially affected in ChL1-2b and ChL3-4, we tested whether the effects were additive by examining the chimera ChL1-2b + L3-4 in which both extracellular regions are swapped (Fig. 6a). The $G$–$V$ curve of the combination chimera was shifted $-34$ mV compared to SlHv1, similar to the $G$–$V$ shift observed with ChL1-2b alone, indicating that the swap in the S1–S2 loop had a dominant effect. Consistent with this observation, the activation of ChL1-2b + L3-4 was accelerated compared to SlHv1 (Fig. 6b, left), and its deactivation was not slowed down (no increase in $t_{1/2}$, Fig. 6b, right). The $G$–$V$ curve of ChL1-2b + L3-4 appeared to span a significantly wider voltage range than the $G$–$V$ curves of the parent proteins AoHv1 and SlHv1. As a result, the chimeric channels started opening within the voltage range of activation of AoHv1 (Fig. 6a, blue-shadowed area), but maximal conductance was reached within the voltage range of activation SlHv1 (Fig. 6a, orange-shadowed area).

Of the chimeras with swapped C-terminal regions, the $G$–$V$ curves of ChCT2 and ChCT3 were similar to the $G$–$V$ curve of SlHv1 ($\Delta V_{1/2}$ within $\pm 5$ mV) with no significant changes in

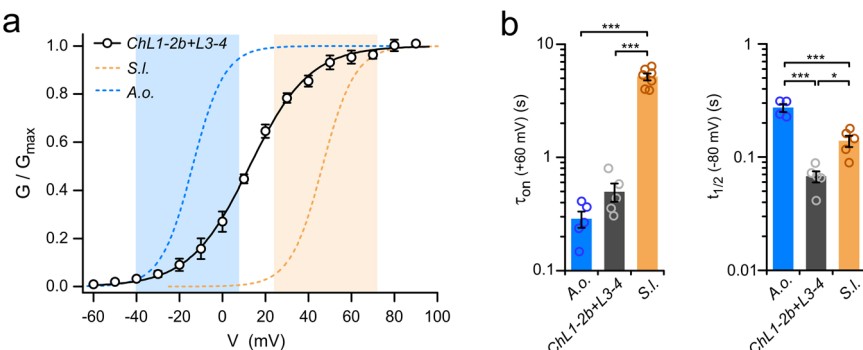

**Fig. 6 Activation properties of Sl/AoHv1 chimeric channel with swaps in S1–S2 and at S3–S4 loops. a** $G$–$V$ curve of ChL1-2b + L3-4 chimera compared to parent channels SlHv1 and AoHv1 (pH$_i$ = pH$_o$ = 6.0). Conductance of chimera starts increasing within the voltage range of AoHv1 activation (blue-shaded area) and reaches maximum within the voltage range of SlHv1 activation (orange-shaded area). $G$–$V$ parameters derived from Boltzmann fit of the data are: $V_{1/2} = 12.7 \pm 1.7$ mV, slope $= 13.9 \pm 1.3$ mV ($n = 5$). **b** Time constant of activation ($\tau_{on}$) and half-maximal deactivation time ($t_{1/2}$) of ChL1-2b + 3-4 compared to parent channels SlHv1 and AoHv1. $\tau_{on}$ was measured at 60 mV and $t_{1/2}$ was measured at $-80$ mV. Bars are mean values from 4 to 5 independent measurements ± SEM. A one-way ANOVA multiple comparison test with Tukey's post hoc correction was used for statistical analysis: *$p < 0.05$, ***$p < 0.001$.

activation rates (Fig. 5b, e). However, the $G$–$V$ curve of ChCT1 was shifted to more positive potentials compared to SlHv1 ($\Delta V_{1/2} = 18$ mV) despite a strong acceleration of channel opening (Fig. 5b, e). Finally, the chimera with the swap within the intracellular S2–S3 loop, ChL2-3, exhibited a $G$–$V$ curve with a large positive shift (Fig. 5e, $\Delta V_{1/2} = 43$ mV) bringing the overall range of modulation of the SlHv1 voltage dependence of activation by peripheral regions to a remarkable 72 mV ($\Delta V_{1/2}$ range between ChL1-2b and ChL2-3).

SlHv1 and AoHv1 differ in their pH dependence under symmetrical conditions. The $G$–$V$ relationship for SlHv1 shifts ~18 mV per pH unit, whereas the $G$–$V$ relationship for AoHv1 is insensitive to pH under the same conditions (Fig. 3j, k). Previous comparative studies on animal Hvs identified a potential pH$_i$ sensor located in the S2–S3 loop[63]. At the corresponding position, SlHv1 contains a 7-residue insertion which is missing in AoHv1 and in the ChL2-3 chimera (which is otherwise identical to SlHv1). Thus, we examined whether the $G$–$V$ curve of the ChL2-3 channel was sensitive to pH under symmetrical conditions. Due to low expression, we were unable to perform the measurements at pH >6.0, but we were able to compare pH 5.5 and 6.0 (Supplementary Fig. 6). We found that, as observed with AoHv1, the $G$–$V$ curve did not shift between pH 5.5 and 6.0 in the chimera, while it shifted ~7.6 mV in SlHv1, under the same conditions (Fig. 3j). These data suggest that the 7-residue insertion in the S2–S3 loop is responsible for the different pH sensitivity between SlHv1 and AoHv1 in the absence of a transmembrane pH gradient.

## Discussion

Electricity is a key element for growth and development in various types of organisms including fungi. Action potential-like spikes, occurring spontaneously or triggered by cyanide, were first reported in the water mold *Neurospora crassa* in the '70s[64,65]. Similar electrical signals were subsequently recorded in other fungi species, from gill mushrooms to filamentous fungi[66,67]. Transcellular electric currents, ubiquitous among mycelial fungi, enter the tips of the hyphae and exits their distal regions, and are critical for hyphal polarization and branching[68–70]. They are also associated with material transfer and hydraulic pressure[71]. These currents have been shown to be carried primarily by protons in *N. crassa* and many other types of fungi[70,72].

Proton transport is essential for fungi beyond the regulation of pH and membrane potential, yet only the H$^+$-ATPase Pma1 from the plasma membrane is clearly described in this context, along with the PacC/Rim signaling transduction pathway[73]. The identification of members of the Hv channel family in all five major phyla of the fungi kingdom (*Ascomycota, Basidiomycota, Chytridiomycota, Zygomycota*, and *Glomeromycota*, Supplementary Fig. 1) suggests that these proteins are ubiquitous components of proton transport mechanisms in fungi. The different biophysical characteristics of SlHv1 and AoHv1, and in particular the large difference in voltage range of activation, provide clues on possible physiological functions of these channels. SlHv1 opens only when the electrochemical gradient favors outward H$^+$ movement and so it is similar in behavior to the large majority of animal Hvs that act as proton extruders and counteract intracellular acidification and membrane depolarization caused by NOX enzymes[12–14]. Conversely, AoHv1 opens when the electrochemical gradient favors inward H$^+$ movement and so it can produce intracellular acidification and membrane depolarization. The only known Hv with similar behavior is the channel from the dinoflagellate *Karlodinium veneficum* (kHv1)[38]. Dinoflagellates are well-known for their ability to emit flashes of light thanks to cytoplasmic

structure called scintillons. Hv channels like kHv1 are believed to be responsible for the initiation of the action potential that triggers the bioluminescent process within the scintillon[74,75]. Similarly, AoHv1 could provide the depolarizing current driving action potential-like spikes in molds[64,66]. However, the unprecedented range of modulation of voltage-dependent activation by the transmembrane pH gradient observed in AoHv1 ($\Delta V_{1/2} = $~90 mV/$\Delta$pH unit, Fig. 3g–i) indicates that the channel can also work as a proton extruder under the appropriate conditions. As a result, some of AoHv1 functions could overlap with those of exclusive proton extruders like SlHv1.

A large number of fungi are pathogenic to human, wildlife, or agricultural products. The phylogenetic tree of Supplementary Fig. 1 includes representatives from *Rhodotorula spp., Basidiobolus spp., Sporothrix spp., Fusarium spp., Absidia spp., Cladophialophora spp., Talaromyces spp.*, and *Thielavia spp.*, which can infect the human skin, respiratory and gastrointestinal tracts, bloodstream, eyes, and brain, as well as representatives from species like *Rhizopus, Sclerotinia*, and *Verticilium*, which are pathogenic to crops and cultivars such as maize, rice, sunflower, canola, and cruciferous vegetables. The ability of fungi to adapt to a wide range of pHs and to actively modify the pH in their surroundings makes them extremely difficult to eliminate. Drugs targeting fungal Hvs could provide new tools to study the functions of these channels in vivo and to fight mycotic infections. Zn$^{2+}$-mediated inhibition has an important role in the physiology of animal Hvs[76] and has been used as a pharmacological tool to study these channels[77]. In contrast, fungal Hvs are particularly resistant to this inhibitor (Fig. 4c–f). The lack of histidine residues corresponding to those proposed to coordinate Zn$^{2+}$ in animal Hvs[9,11] (e.g., H140 and H193 in human Hv1, Supplementary Fig. 2a) is likely to be responsible for the low Zn$^{2+}$ sensitivity of fungal channels. On the other hand, guanidine derivatives, such as 2GBI and ClGBI, are more likely to be useful against fungal Hvs. These compounds were able to substantially inhibit SlHv1 and AoHv1 in the same concentration range used for hHv1 (Fig. 4a, b). The small but significant differences in inhibition observed between the fungal Hvs suggest that the compounds could be further optimized to enhance selectivity. Fungal Hvs are highly conserved between species of an individual genus. As a result, the pharmacological characteristics of AoHv1 described here are likely to be shared by Hvs from other *Aspergillus spp.*, including *Aspergillus flavus*, a well-known human pathogen (AfHv1 differs from AoHv1 only by one amino acid, Supplementary Fig. 1).

Upon membrane depolarization, the VSD of Hv channels undergoes conformational changes that result in gate opening and proton conduction[37,78,79]. Rearrangement in the S1 and S4 helices were shown to play critical roles in this process[79,80]. The activation gate is thought to be located within the transmembrane part of the VSD[33] and little is known about the participation of intra- and extracellular peripheral regions in channel gating. Earlier studies found that the CCD mediates cooperative activation of the two channel subunits[27,28,30], while portions of the N-terminal region and loop connecting S2 and S3 contribute to intracellular pH sensitivity[63,81]. The N-terminal region of human Hv1 was shown to be differentially processed, leading to distinctive internalization between isoforms[82], and to harbor a site for PKC phosphorylation that enhances channel gating[83]. In addition, the N-terminal region and part of S3 from the sea urchin Hv1 were found to accelerate channel activation when co-transplanted to slow-activating mouse Hv1[84]. In other voltage-gated ion channels, the extracellular loops of the VSDs are involved in interactions between channel-forming subunits and auxiliary/regulatory subunits[85] and in the Kv1.2 channel,

the length, and composition of the S3–S4 loop was shown to fine-tune voltage sensitivity[86]. These observations suggest that protein regions beyond the transmembrane portion of the VSD can provide important contributions to Hv channel function.

In this work, we found that intra- and extracellular peripheral regions of fungal Hvs are major determinants of their voltage dependence of activation, as swapping portions of these regions between AoHv1 and SlHv1 produced shifts in the G–V relationships of the chimeric channels of up to ~70 mV (Fig. 5d, e). In particular, the loop connecting S1 to S2 provided a dominant contribution to the difference in kinetics and voltage range of activation between AoHv1 and SlHv1 (Fig. 5d and Supplementary Fig. 2a). The S3–S4 loop provided a smaller contribution, but as a result of a complementary mechanism. While swapping the S1–S2 loop mostly affected the rate of opening (Fig. 5b), swapping the S3–S4 loop mostly affected the rate of closing (Fig. 5c). Transplanting both extracellular loops of AoHv1 into SlHv1 resulted in a chimeric channel that starts opening within the voltage range of activation of AoHv1 and becomes fully open within the voltage range of activation of SlHv1 (Fig. 6). Future studies should investigate whether small molecule compounds or proteins capable of binding the S1–S2 loop of fungal Hvs can shift their voltage range of activation leading to inhibition or enhancement of channel activity.

In animal Hvs, the S1 helix mediates intersubunit interactions that are important for cooperative gating[30,57,78,79]. A cysteine substitution introduced in the S1–S2 loop, close to the outer end of S1 (I127C in hHv1), was shown to form a spontaneous disulfide bond[30], which enhanced allosteric coupling between subunits[57]. Fungal Hvs contain an endogenous cysteine either at, or in proximity of, the position homologous to I127 of hHv1 (Supplementary Fig. 2a), suggesting that the intersubunit interface of these channels extends into the S1–S2 loop. We hypothesize that this interface could engage in intra- or intersubunit interactions with the S3–S4 loop and the S4 helix to set the range of voltage-dependent activation. Future structural and site-directed mutagenesis studies will be needed to test this hypothesis and determine the mechanisms underlying gating modulation by peripheral regions in fungal Hv channels.

## Methods

**Protein sequence analysis**. Multiple sequence alignment and phylogenetic analysis were performed using Clustal Omega from EMBL-EBI tools[87]. Phylogenetic tree and cladogram were constructed with iTOL 5.6.2[88]. Tree scale is at 0.1. Primary sequences for AoHv1 and SlHv1 were further analyzed with MPEx[89] and Coils – ExPASy[90]. The following protein sequences were used to construct the cladogram and the phylogenetic tree, and to search for potential fungal otopetrins.

*Fungal Hvs in the cladogram. Hypsizygus marmoreus* (RDB21275.1, 215aa); *Amanita muscaria* (KIL69657.1, 218aa); *Psilocybe cyanescens* (PPQ83343.1, 214aa); *Suillus luteus* (KIK49332.1, 223aa); *Scleroderma citrinum* (KIM55885.1, 225aa); *Galerina marginata* (KDR81513.1, 217aa); *Mycena chlorophos* (GAT47218.1, 202aa); *Agaricus bisporus* (XP_007326257.1, 183aa); *Piriformospora indica* (CCA68166.1, 210aa); *Fusarium oxysporum* (XP_031056756.1, 230aa); *Sclerotinia sclerotiorum* (XP_001595616.1, 226aa); *Cladophialophora immunda* (XP_016251813.1, 259aa); *Talaromyces marneffei* (EEA28233.1, 309aa); *Penicillium brasilianum* (CEJ60805.1, 205aa); *Aspergillus oryzae* (XP_001825565.1, 211aa); *Aspergillus flavus* (GenBank: XP_002381556.1, 211aa).

*Additional Hvs included in the phylogenetic tree. Rhodotorula toruloides* (EGU12623.1, 262aa); *Spizellomyces punctatus* (XP_016610604.1, 227aa); *Lobosporangium transversale* (XP_021881983.1, 208aa); *Mortierella elongata* (OAQ32698.1, 206aa); *Basidiobolus meristosporus* (ORX99742.1, 207aa); *Rhizophagus clarus* (GBC03452.1, 235aa); *Bifiguratus adelaidae* (OZJ02879.1, 252aa); *Absidia repens* (ORZ16286.1, 220aa); *Rhizopus microspores* (CEI92734.1, 204aa); *Tieghemostelium lacteum* (KYQ94119.1, 262aa); *Polysphondylium violaceum* (KAF2071235.1, 331aa).

*Hvs listed as references from other organisms (cladogram and phylogenetic tree). Nicoletia phytophile* (AMK01488.1, 239aa); *Octopus bimaculoides* (XP_014789275.1, 348aa); *Ciona intestinalis* (NP_001071937.1, 342aa); *Danio rerio* (NP_001002346.1, 235aa); *Xenopus tropicalis* (NP_001011262.1, 230aa); *Homo sapiens* (NP_001035196.1, 273aa); *Mus musculus* (NP_001035954.1, 269aa); *Gallus gallus* (NP_001025834.1, 235aa); *Alligator sinensis* (XP_006015244.1, 239aa).

*Search for otopetrin orthologs.* Otopetrin proteins from human (NP_819056.1, NP_835454.1, NP_001258934.1, NP_839947.1), zebrafish (NP_942098.1), frog (XP_012811170.1), fruitfly (NP_001259255.1, NP_722888.1), and nematode (XP_001672406.1) were used in BLAST search for identification of possible homologs in fungi and returned with no hits.

**Channel expression in *Xenopus* oocytes**. DNA constructs encoding wild-type SlHv1 and AoHv1 and chimeras ChL1-2 and ChCT3 were synthesized by Gen-Script after codon optimization for protein expression in animal cells. A construct containing the cDNA sequence from *HVCN1* in pGEMHE[91] was used to express human Hv1[33]. Chimeras ChL1-2a, ChL1-2b, ChL2-3, ChL3-4, ChCT1, and ChCT2 were prepared using standard PCR techniques. All constructs were generated by subcloning the sequences flanking BamHI/XbaI in the pGEMHE[91] vector and linearized with NheI or SphI restriction enzymes (New England Biolabs) before in vitro transcription. mRNAs were synthesized using T7 mMessage mMachine transcription kit (Ambion) or HiScribe™ T7 ARCA mRNA Kit (with tailing) (New England Biolabs). All constructs were confirmed by sequencing, and RNA quality was tested by agarose gel electrophoresis. *Xenopus* oocytes from Ecocyte Bioscience or Xenopus 1 were injected with mRNAs (50 nl per cell, 0.5–1.5 ng/nl) 1–3 days before the electrophysiological measurements. Injections were performed with a Nanoject II (Drummond Scientific). Cells were kept at 18 °C in ND96 medium containing 96 mM NaCl, 2 mM KCl, 1.8 mM CaCl$_2$, 1 mM MgCl$_2$, 10 mM HEPES, 5 mM pyruvate, 100 mg/ml gentamycin (pH 7.2).

**Patch-clamp measurements**. Voltage-clamp measurements were carried out either in inside-out patch or outside-out configurations, using an Axopatch 200B amplifier controlled by pClamp10 software through an Axon Digidata 1440A (Molecular Devices). The signal was lowpass filtered at 1 kHz (Bessel, −80 dB/decade) before digitalization (2 kHz sampling). Inside-out patch-clamp experiments were performed under various pH conditions as specified in main text and figures. Bath or pipette recording solution at pH 6.0 contained 100 mM 2-(N-morpholino)ethanesulphonic acid (MES), 30 mM tetraethylammonium (TEA) methanesulfonate, 5 mM TEA chloride, 5 mM ethylene glycol-bis(2-aminoethyl)-N,N,N′,N′-tetra-acetic acid (EGTA), adjusted to pH 6.0 with TEA hydroxide. Recording solution at pH 5.5 contained 100 mM MES, 50 mM TEA methanesulfonate, 5 mM TEA chloride, adjusted with TEA hydroxide; solution at pH 6.5 contained 100 mM 1,4-piperazinediethanesulfonic acid, 5 mM TEA chloride, adjusted with TEA hydroxide. Outside-out measurements were performed in asymmetrical pH condition (pH$_i$ = 6.0, pH$_o$ = 7.0). Solution at pH 7.0 contained 100 mM 3-(N-morpholino)propanesulfonic acid, 40 mM TEA methanesulfonate, 5 mM TEA chloride with or without ZnCl$_2$ at final concentrations indicated in the text. All tested compounds were at the highest purity commercially available. Intracellular inhibitors 2-guanidinobenzimidazole (2GBI) and 5-chloro-2-guanidinobenzimidazole (ClGBI) were from Sigma-Aldrich. All measurements were carried out at 22 ± 1 °C. Pipettes had 1–3 MΩ access resistance. Unless otherwise specified, the holding potential was either −60 or −80 mV. Channel inhibition was determined by isochronal current measurements at the end of the depolarization pulses. For mechanical stimulation of membrane patches, a high-speed pressure clamp (HSPC-1, ALA Scientific) controlled by pCLAMP 10.2 was used to apply negative pressure pulses through the patch pipette.

Gating charges were estimated using the limiting slope method as previously described[45,46]. For SlHv1, voltage ramps from −80 to −20 mV with rates of 0.2 or 0.5 mV/s were used. To accelerate channel activation, pH$_i$ was 5.5 and pH$_o$ was 6.0. For AoHv1, voltage ramps from −80 to −10 mV with rates of 0.5 or 1 mV/s were used. The activation of this channel was fast enough to conduct the measurements at pH$_i$ = pH$_o$ = 6.0. No significant differences were observed between ramps at different rates for each channel. The voltage protocol included a pre-pulse sufficiently positive to reach maximal conductance. Pre-pulse voltage was 30 mV for SlHv1 (asymmetrical pH conditions) and 60 mV for AoHv1 (symmetrical pH conditions).

**Homology modeling and similarity score mapping**. The homology model comprising the VSD and CCD of SlHv1 (Fig. 5a and Supplementary Fig. 5—Model 1) was generated using I-TASSER[92] and Swiss-pdb Viewer[93] based on the crystal structures of mHv1cc (PDB 3WKV), and the isolated CCD from mouse Hv1 (PDB 3VMX)[25,32]. The homology model of the SlHv1 VSD based on CiVSP (Supplementary Fig. 5—Model 2) was generated using the same approach and the crystal structure of the VSD of CiVSP (PDB 4G80)[61] as a template. The sequences of SlHv1 and AoHv1 covering the homology model structure (from S0 to the C-terminus) were aligned using Clustal Omega[87] with modifications aimed at preserving register of structural domains. Specifically, when shortening the SlHv1 CCD to match AoHv1, individual heptad repeats were removed keeping the register of the remaining repeats unaltered. The similarity

scores were determined based on Blocks Substitution Matrix 62 (BLOSUM62) and affine gap penalties (opening: 8, extension: 1). The scores were then normalized and used to derive the scale indicating the deviation from average similarity. The scale (smoothed by adjacent averaging on a five-residue window and ranging from 0 to 1) was converted into a color gradient scale (blue to red) and mapped on the SlHv1 homology models in PyMOL (Schrödinger LLC). Predictions of CCD regions were performed with COILS (https://embnet.vital-it.ch/software/COILS_form.html), guided by structural information available for CCDs of animal Hvs.

**Data analysis**. Current traces were analyzed using Clampfit10.2 (Molecular Devices) and Origin8.1 (OriginLab). Leak subtraction, rundown correction, and assessment of current inhibition were carried out as previously described[33]. Derivation of $G$–$V$ relationships from $I$–$V$ curves was performed using equation:

$$G(V) = I(V)/(V - V_{rev}) \qquad (1)$$

where $V_{rev}$ is the reversal potential of the current. Due to the high $H^+$ selectivity of Hv channels, $V_{rev} \approx E_H$. $G(V)$ values were then divided by $G_{max}$ for normalization. $G$–$V$ relationships were also derived from tail currents, as described in earlier work[42]. Current rundown was corrected using a reference depolarization step preceding the test depolarization. $G$–$V$ plots were fitted with the Boltzmann equation:

$$G/G_{max} = 1/\left(1 + \exp\left(V_{1/2} - V\right)/s\right) \qquad (2)$$

where $V_{1/2}$ is the potential of half-maximal activation, and $s$ is the slope parameter. $\tau_{on}(V)$ values were calculated by fitting currents traces in response to depolarizing voltage steps with the single-exponential equation:

$$I(V, \ t) = a \cdot \exp(-t)/\tau_{on}(V) + c \qquad (3)$$

Half deactivation times ($t_{1/2}(V)$) were measured by calculating the time the tail currents took to decay to $I_o(V)/2$, where $I_o(V)$ is the current at the beginning of the repolarization step. The effective gating charge ($z_g$) associated with activation of SlHv1 and AoHv1 was estimated from the linear fit of the logarithm of the open probability [ln($P_o$)] as a function of voltage under conditions in which $P_o$ is very low (limiting slope method[45,46]). $P_o$ was measured as $G/G_{max}$ and $z_g$ was derived from the *slope* of the linear fit through the equation:

$$z_g = (k_B T/e_o) \cdot slope \qquad (4)$$

where $k_B$ is the Boltzmann constant, $T$ is the absolute temperature, and $e_o$ is the elementary charge.

**Statistics and reproducibility**. All statistical analysis was performed using OrginLab 8.1 (OriginLab). Data are represented as mean ± SEM, unless otherwise indicated. Datasets with two conditions were compared by applying a Welch's *t*-test. Datasets containing more than two conditions were compared using one-way ANOVA test with Tukey's post hoc correction.

**Reporting summary**. Further information on research design is available in the Nature Research Reporting Summary linked to this article.

## Data availability
Examples of current traces used for analysis are provided in Figs. 1–4, and Supplementary Figs. 3–4. Individual data points for all figures are provided as Supplementary Data 1. Other data and materials are available upon reasonable request.

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

## Acknowledgements

The authors wish to thank Dr. Peng Chen for advice on similarity score mapping and members of the Tombola lab for useful feedback on the manuscript. This work was supported by the National Institute of General Medical Sciences, through grant R01GM098973 to F.T. The authors thank the Chao Family Comprehensive Cancer Center at the University of California, Irvine for access to shared resources supported by the National Cancer Institute under award P30CA062203.

## Author contributions

F.T. conceived the idea and oversaw the project; C.Z. and F.T. designed experiments; C.Z. performed experiments; C.Z. and F.T. analyzed data; C.Z. and F.T. wrote the manuscript.

## Competing interests

The authors declare no competing interests.
