## [Peer Review File · Communications Biology]

Reviewers' comments:

Reviewer #1 (Remarks to the Author):

The authors have identified a number of sequences similar to voltage-gated proton (Hv) channels in fungi. When testing two of these fungi Hv channels they find that these are bona fide Hv channels, however, with different pH dependence, different voltage dependence, and different pharmacology than mammalian hv1 channels. The authors also test different chimeras of fungi Hv channels and find that the S1-s2 loop and the connector between S4 and the C-terminus play important roles in determining the range of voltage activation. The data is carefully collected and the conclusions are reasonable. The findings will be interesting for a larger community to understand how voltage activates this class of ion channels. I have only some minor comments.

1. Pg. 2. "Channels form" should be form.

2. Pg. 7. "AoHv1 carried massive inward currents when pHi was lower than pHo" I think the pH are reversed?

3. Pg. 13. "our data suggest that small molecule compounds or proteins capable of binding the S1-S2 loop of fungal Hvs could shift their voltage range of activation". Not sure the authors showed this. Seems like a stretch of their findings...

Reviewer #2 (Remarks to the Author):

This paper explores the characteristics of fungal Hv1's with an eye towards using Hv1 as a drug target. The experiments are well designed with appropriate conditions and controls and the analysis and conclusions are well founded. Hv1 represents a novel fungal target and provides an additional tool towards understanding of important cellular processes in fungi, making this paper of wide interest and potential importance.

There are some points that the authors should address:

1) The rationale for choice of non-fungal sequences included in the cladogram and phylogenetic tree should be explained.

2) The homology model was based on available crystal structures of Hv1; however, there is substantial experimental evidence showing that the mouse Hv1 VSD crystal structure, which is of a chimeral construct, probably does not accurately represent the native structure of Hv1. The authors should make a homology model based on the crystal structure(s) of the close homolog CiVSP and identify any differences that would substantially change the identities of the chimeras made. I stress that experiments with new chimeras based on the other model are not necessary.

3) One of the main differences identified by the authors between the two different fungal Hv1's studied is an unusual 7 residue insertion in the loop between S2-S3 of the basidiomycete fungi shown, that does not appear in the ascomycete sequences nor in animals. The pH dependence of these two Hv1's is quite different, but the authors do not explicitly address whether this reflects changes in pHi vs pHo sensing. As this region has been shown to have a role in pHi sensing in animals, this point should be addressed explicitly. In particular, effects of absolute pH (rather than delta pH) should be made explicit.

4) Taking this S2-S3 insertion out of *S. luteus* exaggerates its difference from *A. oryzae*, moving the G-V curve more to the right. *S. luteus* also has an insertion at the beginning of the C-terminus compared to *A. oryzae*, removal of which moves the G-V curve in the same direction. What does the

double chimera ChCT1-ChS2-3 do? This seems like an obvious experiment that goes along with effects of the double chimeras of the external interhelical loops.

Reviewer #3 (Remarks to the Author):

Dr. Tombola's group discovered novel voltage-gated proton channel, Hv1, from fungi genomes and characterized electrophysiological and pharmacological properties in *Xenopus* oocyte expression. Two fungal Hv1 orthologs showed distinct voltage-dependence and gating kinetics as well as distinct sensitivity to Hv1 inhibitors. Then they analyzed regions that make these differences to find that non-transmembrane regions play some role in channel gating. Results are convincing, techniques are sound and the paper is well written. Although this paper is rather descriptive, the discovery of fungal Hv1 is remarkable and the paper includes some novel and interesting findings on molecular mechanisms of Hv1 which will merit publication. I have following concerns which would be carefully addressed.

1. Both of SI and Ao Hv1 have coiled-coil domain in the C-terminal cytoplasmic region. CCD was reported to facilitate dimer formation of Hv1. Are fungal Hv1s are dimers? To address this point, it will be helpful to test if rising phase of current activation is sigmoidal, since previous works have shown that sigmoidal pattern is one of hallmark features of dimer cooperativity of channel gating. Showing other information such as western blot data showing shift of molecular weight to the dimer in the presence of cross linker molecule or evidence by FRET measurement will also be helpful.
2. Fungi is engulfed by phagocytes which also have Hv1. Do Hv1 ortholog gene exist pathogenic fungi such as *Candida* or *Aspergillus*? This will be an important issue in medical science.
3. Possible functions of "peripheral" regions in channel activation are interesting. However, there is no mechanistic insight into these regions. For example, S1-S2 extracellular region may be involved in dimer interaction, since several previous papers showed residues in S1 play important roles in dimer cooperativity. Molecular mechanisms should be discussed in more detail.

I also have following minor specific comments.

4. I am not sure whether readers could understand "peripheral" regions as the meaning of non-transmembrane regions of Hv1. Do you mean "peripheral" by both extracellular and intracellular regions? Please see other examples of usage of the word "peripheral region" in other ion channel researches. "Peripheral" regions will give an impression that Hv1 has auxiliary subunit, as in ionotropic glutamate receptor, or K(ATP) channel which surrounds central channel forming subunit. It could be rephrased, for example, "nontransmembrane regions"?
5. In page 7, it is said "AoHv1 did meet this expectation, with little change in $V_{1/2}$ within the 5.5-6.5 pH range. However, SIHv1 showed a clear pH dependence within the same pH range, with a $\Delta V_{1/2}$ of 20 mV per pH unit." Authors would discuss possible molecular mechanisms underlying this difference between the two fungal orthologs.
6. I could not understand what the following sentence means; (4-5th line on page 8) "Overall AoHv1 was the least mechanosensitive of the compared channels, which is in agreement with its voltage dependence and kinetics of activation". Readers may not understand relationship between "voltage dependence and kinetics of activation" and "mechanosensitive" of Hv1.
7. In Fig1e, slope of G-V curve seems steeper in hHv1 than fungal Hv1. Is it significant, or could be due to some error by distinct current density in excised out patch membrane?
8. Other papers would be cited in discussion for highlighting novel points in the studies of chimeric fungal Hv1s.

Some possible roles of so called "peripheral" (cytoplasmic) regions in channel properties of Hv1 have

been previously described (but without clear mechanisms revealed).

Berger, T.K. et al, *J Physiol* 595, 1533-1546 (2017) described some role of N-terminus in pH-dependent gating of mammalian Hv1s. Sakata, S. et al, *Biochem. Biophys. Acta.* 1858(12):2972-2983 (2016) described some role of N-terminal region in gating kinetics. N-terminal region is the site for PKC phosphorylation, leading to some minor channel properties (Musset B, et al. *J Biol Chem.* 285(8):5117-21(2010)). N-terminal region of mammalian Hv1 is differentially processed (or initiator methionine is different) with distinct current density and is related to human B-lymphocyte tumor (Hondares E, et al. *Proc Natl Acad Sci U S A.* 111(50):18078-83(2014)).

9. In FigS2, only 8 amino acids of CCD are shown and some orthologs contain many serial glutamic acids, which is atypical for CCD. Coiled coil nature can easily be predicted by web server. Although CCD is not the main story of this paper, longer region of CCD would be shown in this figure.

.

We would like to thank the Reviewers for their helpful assessment of our manuscript. We provide detailed answers to their remarks below (text in blue). Major revisions of the text (manuscript + SI) are colored in red in the marked-up file.

Reviewer #1:

The authors have identified a number of sequences similar to voltage-gated proton (Hv) channels in fungi. When testing two of these fungi Hv channels they find that these are bona fide Hv channels, however, with different pH dependence, different voltage dependence, and different pharmacology than mammalian hv1 channels. The authors also test different chimeras of fungi Hv channels and find that the S1-s2 loop and the connector between S4 and the C-terminus play important roles in determining the range of voltage activation. The data is carefully collected and the conclusions are reasonable. The findings will be interesting for a larger community to understand how voltage activates this class of ion channels. I have only some minor comments.

1. Pg. 2. "Channels form" should be form.

Done.

2. Pg. 7. "AoHv1 carried massive inward currents when pHi was lower than pHo" I think the pH are reversed? AoHv1 does conduct large inward currents when pHi is lower than pHo. This is because, despite the unfavorable chemical gradient, the overall electrochemical gradient drives an inward proton flow. The counterintuitive property is due to the combination of two factors: 1) AoHv1 activates within a voltage range that is more negative than the Nernst potential for protons, and 2) the channel G-V curve displays unusually large negative shifts in response to positive transmembrane pH gradients (pHo-pHi > 0).

3. Pg. 13. "our data suggest that small molecule compounds or proteins capable of binding the S1-S2 loop of fungal Hvs could shift their voltage range of activation". Not sure the authors showed this. Seems like a stretch of their findings...

We agree with the reviewer and changed the sentence to a call for future studies to investigate whether small molecule compounds or proteins capable of binding the S1-S2 loop of fungal Hvs can shift their voltage range of activation.

Reviewer #2:

This paper explores the characteristics of fungal Hv1's with an eye towards using Hv1 as a drug target. The experiments are well designed with appropriate conditions and controls and the analysis and conclusions are well founded. Hv1 represents a novel fungal target and provides an additional tool towards understanding of important cellular processes in fungi, making this paper of wide interest and potential importance.

There are some points that the authors should address:

1) *The rationale for choice of non-fungal sequences included in the cladogram and phylogenetic tree should be explained.*

We now provide information on the non-fungal sequences included in the cladogram and phylogenetic tree on page 5 and legend of Fig. S1. We also provide a reference to a more detailed phylogenetic analysis of animal Hvs (Smith et al. *PNAS* 2011). In the previous version of the manuscript, some confusion might have arisen from the fact that, in addition to animal Hvs, we included species from slime molds in Fig. S1. We did this because these organisms used to be considered part of the *Fungi* kingdom due to their "fungus-like" appearance, but they were later reclassified as protists. Our phylogenetic analysis indicates that the sequences of Hvs from slime molds are more similar to animal Hvs than fungal Hvs. We have clarified this in the revised text.

2) The homology model was based on available crystal structures of Hv1; however, there is substantial experimental evidence showing that the mouse Hv1 VSD crystal structure, which is of a chimeral construct, probably does not accurately represent the native structure of Hv1. The authors should make a homology model based on the crystal structure(s) of the close homolog CiVSP and identify any differences that would substantially change the identities of the chimeras made. I stress that experiments with new chimeras based on the other model are not necessary.

To check the extent to which the sequence divergence patterns used to target our chimeras depend on the modeling template, we generated an additional homology model for SlHv1 based on the structure of the VSD of CiVSP (Li et al. *Nat. Struct. Mol. Biol.* 2014). The scale of sequence divergence between SlHv1 and AoHv1 was mapped also on this model. The new Fig. S5 shows a comparison between the models based on mHv1cc and CiVSP (Model 1 and 2, respectively). We found local structural differences in all the divergent regions and, in particular, in the region containing the S2-S3 loop. However, these differences did not substantially change the identities of the chimeras made. We provide information on the differences between Model 1 and 2 on page 10 and in the legend of Fig. S5.

3) One of the main differences identified by the authors between the two different fungal Hv1's studied is an unusual 7 residue insertion in the loop between S2-S3 of the basidiomycete fungi shown, that does not appear in the ascomycete sequences nor in animals. The pH dependence of these two Hv1's is quite different, but the authors do not explicitly address whether this reflects changes in pHi vs pHo sensing. As this region has been shown to have a role in pHi sensing in animals, this point should be addressed explicitly. In particular, effects of absolute pH (rather than delta pH) should be made explicit.

We measured the G-V curve of ChL2-3 chimera (which lacks the 7-residue insertion found in SlHv1) under symmetrical pH 5.5 and 6.0, and compared the shift in $V_{1/2}$ to the corresponding shifts observed in SlHv1 and AoHv1. The data is presented in the new Fig. S6. As indicated by the reviewer, the 7-residue insertion of SlHv1 is located at a position that was previously found to contain a pHi sensor in animal Hvs. We find that deleting the insertion remove pH sensitivity under symmetrical conditions, making the ChL2-3 chimera more similar to AoHv1 which naturally lacks the insertion. The new findings are presented on page 12.

4) Taking this S2-S3 insertion out of *S. luteus* exaggerates its difference from *A. oryzae*, moving the G-V curve more to the right. *S. luteus* also has an insertion at the beginning of the C-terminus compared to *A. oryzae*, removal of which moves the G-V curve in the same direction. What does the double chimera ChCT1-ChS2-3 do? This seems like an obvious experiment that goes along with effects of the double chimeras of the external interhelical loops.

As suggested by the reviewer, we generated the double chimera ChCT1-ChL2-3 and attempted to measure its G-V curve. However, we were unsuccessful due to the low expression level of this channel and the high voltages required for the measurements. By the time we could detect measurable currents, the membrane was too unstable to sustain strong depolarizations.

Reviewer #3:

Dr. Tombola's group discovered novel voltage-gated proton channel, Hv1, from fungi genomes and characterized electrophysiological and pharmacological properties in *Xenopus* oocyte expression. Two fungal Hv1 orthologs showed distinct voltage-dependence and gating kinetics as well as distinct sensitivity to Hv1 inhibitors. Then they analyzed regions that make these differences to find that non-transmembrane regions play some role in channel gating. Results are convincing, techniques are sound and the paper is well written. Although this paper is rather descriptive, the discovery of fungal Hv1 is remarkable and the paper includes some novel and interesting findings on molecular mechanisms of Hv1 which will merit publication. I have following concerns which would be carefully addressed.

1. Both of Sl and Ao Hv1 have coiled-coil domain in the C-terminal cytoplasmic region. CCD was reported to facilitate dimer formation of Hv1. Are fungal Hv1s dimers? To address this point, it will be helpful to test if rising phase of current activation is sigmoidal, since previous works have shown that sigmoidal pattern is one of hallmark features of dimer cooperativity of channel gating. Showing other information such as western blot data showing shift of molecular weight to the dimer in the presence of cross linker molecule or evidence by FRET measurement will also be helpful.

We now report that the rising phases of current activation for both SlHv1 and AoHv1 are sigmoidal (Fig. S3a-b) and refer to earlier work on animal Hvs showing that the sigmoidal pattern is an indication of cooperative gating in the dimer (Fujiwara et al. *J. Physiol.* 2012). In addition, we explored the cooperativity of gating by measuring the gating charge (z_g) associated with channel opening using the limiting slope method, as other groups previously did with animal Hvs (Gonzalez et al. *Nat. Struct. Mol. Biol.* 2010; Fujiwara et al. *Nat. Commun.* 2012). We calculated z_g values of approximately 5 for both SlHv1 and AoHv1 (Fig. S3c-e). Since the expected contribution to z_g of each subunit is up to 3 charges, the new data confirm that the gating of fungal Hvs involves more than one subunit and is cooperative as in animal Hvs. These findings are presented on page 6 of the revised manuscript.

2. Fungi is engulfed by phagocytes which also have Hv1. Do Hv1 ortholog gene exist pathogenic fungi such as *Candida* or *Aspergillus*? This will be an important issue in medical science.

Yes, Hvs orthologs exist in pathogenic fungi as well. In the phylogeny tree of Fig. S1, we included representatives from *Rhodotorula spp.*, *Basidiobolus spp.*, *Sporothrix spp.*, *Fusarium spp.*, *Absidia spp.*, *Cladophialophora spp.*, *Talaromyces spp.* and *Thielavia spp.*, which can infect the human skin, respiratory and gastrointestinal tracts, bloodstream, eyes, and brain. This is clarified in the revised discussion (pages 13-14). Furthermore, we now specify that AoHv1 is highly homologous to the Hvs of other *Aspergillus spp.*, including *Aspergillus flavus*, a well-known human pathogen (AfHv1 differs from AoHv1 only by one amino acid, Fig. S1).

3. Possible functions of “peripheral” regions in channel activation are interesting. However, there is no mechanistic insight into these regions. For example, S1-S2 extracellular region may be involved in dimer interaction, since several previous papers showed residues in S1 play important roles in dimer cooperativity. Molecular mechanisms should be discussed in more detail.

We agree with the reviewer that the S1-S2 extracellular region of fungal Hvs is likely involved in dimer interactions. We now discuss this possibility in the last paragraph of page 15.

I also have following minor specific comments.

4. I am not sure whether readers could understand “peripheral” regions as the meaning of non-transmembrane regions of Hv1. Do you mean “peripheral” by both extracellular and intracellular regions? Please see other examples of usage of the word “peripheral region” in other ion channel researches. “Peripheral” regions will give an impression that Hv1 has auxiliary subunit, as in ionotropic glutamate receptor, or K(ATP) channel which surrounds central channel forming subunit. It could be rephrased, for example, “nontransmembrane regions”?

The protein regions that we identify as important for channel gating are located on the membrane surface rather than the transmembrane region, so we refer to them as “peripheral”. In this context, the term “peripheral” is used with the traditional meaning applied to membrane proteins in general, not to specific ion channels, as peripheral membrane proteins interact primarily with the periphery of the membrane (its surface), whereas transmembrane proteins interact also with its hydrophobic core. To avoid possible confusion, we now specify the meaning of “peripheral” in the introduction (page 4).

5. In page 7, it is said “AoHv1 did meet this expectation, with little change in $V_{1/2}$ within the 5.5-6.5 pH range. However, SIHv1 showed a clear pH dependence within the same pH range, with a $\Delta V_{1/2}$ of 20 mV per pH unit.” Authors would discuss possible molecular mechanisms underlying this difference between the two fungal orthologs.

Following a similar suggestion from reviewer 2 (point 3), we performed measurements to test whether the region between S2 and S3 is involved in the difference in pH dependence under symmetrical conditions between AoHv1 and SIHv1. Our findings are in agreement with previous studies on the role of the S2-S3 loop in pH sensing in animal Hvs and we now address this point on page 12.

6. I could not understand what the following sentence means; (4-5th line on page 8) “Overall AoHv1 was the least mechanosensitive of the compared channels, which is in agreement with its voltage dependence and kinetics of activation”. Readers may not understand relationship between “voltage dependence and kinetics of activation” and “mechanosensitive” of Hv1.

We meant that AoHv1 opens readily in response to membrane depolarization even in the absence of a mechanical stimulus, so it is not surprising that the increase in membrane tension does not make the channel much easier to open. We reworded the sentence eliminating the potentially confusing words “the least mechanosensitive”.

7. In Fig1e, slope of G-V curve seems steeper in hHv1 than fungal Hv1. Is it significant, or could be due to some error by distinct current density in excised out patch membrane?

The slopes of the G-V curves for fungal Hv1s are actually slightly steeper than the G-V for hHv1. We do not think these are significant differences as we measured the gating charges associated with the activation of SIHv1 and AoHv1 using the limiting slope method and obtained values comparable with those previously reported for animal Hvs (see answer to point 1).

8. Other papers would be cited in discussion for highlighting novel points in the studies of chimeric fungal Hv1s. Some possible roles of so called “peripheral” (cytoplasmic) regions in channel properties of Hv1 have been previously described (but without clear mechanisms revealed). Berger, T.K. et al, *J Physiol* 595, 1533-1546 (2017) described some role of N-terminus in pH-dependent gating of mammalian Hv1s. Sakata, S. et al, *Biochem. Biophys. Acta.* 1858(12):2972-2983 (2016) described some role of N-terminal region in gating kinetics. N-terminal region is the site for PKC phosphorylation, leading to some minor channel properties (Musset B, et al. *J Biol Chem.* 285(8):5117-21(2010)). N-terminal region of mammalian Hv1 is differentially processed (or initiator methionine is different) with distinct current density and is related to human B-lymphocyte tumor (Hondares E, et al. *Proc Natl Acad Sci U S A.* 111(50):18078-83(2014)).

We added the suggested references to the revised discussion (page 14).

9. In FigS2, only 8 amino acids of CCD are shown and some orthologs contain many serial glutamic acids, which is atypical for CCD. Coiled coil nature can easily be predicted by web server. Although CCD is not the main story of this paper, longer region of CCD would be shown in this figure.

We now show the predicted CCDs of SIHv1 and AoHv1 compared to the CCD of human Hv1 in Fig. S2b. Our predictions are instructed by both web servers and the crystal structures of corresponding regions of animal Hvs (e.g., 3VMX).

REVIEWERS' COMMENTS:

Reviewer #2 (Remarks to the Author):

My comments have been fully addressed and I recommend publication.

Reviewer #3 (Remarks to the Author):

The authors have addressed all concerns very carefully. I am fully satisfied by this new version. Congratulations on this nice piece of work on the proton channel and fungal biology!